# Effect of 0.8 at.% H on the Mechanical Properties and Microstructure Evolution of a Ti–45Al–9Nb Alloy Under Uniaxial Tension at High Temperature

**Qiqi Yu [1], Daosheng Wen [1,\*], Shouren Wang [1,\*], Beibei Kong [2], Shuxu Wu [1] and Teng Xiao [1]**

[1] School of Mechanical Engineering, University of Jinan, Jinan 250022, China;
20172120498@mail.ujn.edu.cn (Q.Y.); 20172120503@mail.ujn.edu.cn (S.W.);
201821200544@mail.ujn.edu.cn (T.X.)

[2] Department of Mechanical Engineering, Shandong Jiaotong University, Jinan 250022, China;
kongbeibei@sdjtu.edu.cn

\* Correspondence: me_wends@ujn.edu.cn (D.W.); me_wangsr@ujn.edu.cn (S.W.)

**Abstract:** To investigate the effect of hydrogen on the high-temperature deformation behaviors of TiAl-based alloys, the high-temperature tensile experiment was carried out on a Ti–45Al–9Nb (at.%) alloy with the H content of 0 and 0.8 at.%, respectively. Then, the effect of hydrogen on the high-temperature mechanical properties of the as-cast alloy was studied, the constitutive relations among stress, temperature, and strain rate were established, and the microstructure was analyzed. The results indicated that, compared with the unhydrogenated alloy, the flow stress of the hydrogenated alloy was significantly reduced, and the peak stress of the hydrogenated alloy decreased by $(16.28 \pm 0.17)\%$ deformed at $1150\,^\circ\text{C}/0.0004\,\text{s}^{-1}$. Due to the presence of hydride $(\text{TiAl})\text{H}_x$ in the alloy, the elongation showed a decline trend with increasing strain rate at the same deformation temperature. Compared with the unhydrogenated alloy, the elongation of the hydrogenated alloy reduced by $(26.05 \pm 0.45)\%$ $(0.0004\,\text{s}^{-1})$, $(23.49 \pm 0.38)\%$ $(0.001\,\text{s}^{-1})$, and $(14.23 \pm 0.19)\%$ $(0.0025\,\text{s}^{-1})$, respectively, indicating that 0.8 at.% H softened the Ti–45Al–9Nb alloy and reduced the high-temperature plastic deformability. Under the same deformation condition, the deformation extent of the hydrogenated alloy was less than that of the unhydrogenated alloy. There were more residual lamellae in the hydrogenated alloy, and the extent of dynamic recrystallization was lower than that of the unhydrogenated alloy.

**Keywords:** TiAl-based alloys; hydrogen-induced softening; dynamic recrystallization; cracking

## 1. Introduction

TiAl-based alloys are characterized by low density, high specific strength, excellent oxidation resistance, and creep resistance at high-temperature, so it is considered as one of the most promising high-temperature lightweight structural materials used in key components such as aerospace aircraft and automobile engines [1–3]. Due to the poor plasticity of TiAl-based alloys at room temperature, it is difficult to process and deform at room temperature. Furthermore, TiAl-based alloys have a high flow stress even in the thermal deformation process, so high performance dies and equipment are required in plastic forming [4,5]. Therefore, this requires that the dies and equipment can bear a higher load at temperatures of more than 1000 °C, and can work at a high-temperature for a long time, which virtually increases the cost and affects its practical process. Hence, these problems have become serious restrictions in the application of TiAl-based alloys.

In order to reduce the flow stress in the hot working process of TiAl-based alloys, researchers have mainly adopted an alloy composition design and microstructure control [6,7]. A large number of

studies have proven that the addition of the alloying element Mo to TiAl-based alloys is an effective method to reduce the flow stress of TiAl-based alloys [8]. Godor et al. [9] studied the high-temperature deformation behavior of a high Mo–TiAl alloy, and found that the true stress–strain curve of the Ti–45Al–3Mo–0.5Si–0.1B alloy presented typical dynamic recrystallization softening characteristics. Based on the actual compression results, it was found that TiAl-based alloys had a relatively low flow stress and excellent thermal processing performance under the conditions of higher than 1100 °C and lower than 0.01 s$^{-1}$. However, the density of Mo is very large (i.e., 10.22 g/cm$^3$, almost twice as that of Ti), and Mo addition will greatly increases the density of the alloy. Therefore, a way needs to be found that can reduce the flow stress of TiAl-based alloys and does not increase the density. Thermohydrogen treatment (THT) (hydrogenation–hot working–vacuum dehydrogenation) takes hydrogen as a temporary alloying element. It reduces the flow stress of TiAl-based alloys without increasing density, and provides a new way to promote the practical application of TiAl-based alloys [10]. At present, Russia has successfully applied this technology in the production process of BT30 alloy bracket nuts and BT16 alloy large-diameter bolts [11]. Senkov et al. [12] studied the effect of hydrogen on the high-temperature mechanical behavior of a Ti–6Al–4V alloy and found that the peak stress of the alloy with a hydrogen content of 0.4 wt.% decreased by 70% when compared with the unhydrogenated one. Wen et al. [13] studied the influence of hydrogen on the high-temperature deformation behavior of the Ti–46Al–2V–1Cr–0.3Ni alloy and Ti–45Al–5Nb–0.8Mo–0.3Y alloy, and their maximum hydrogen absorption capability was 0.8 and 1.5 at.%, respectively. The results indicated that the peak stress of the Ti–46Al–2V–1Cr–0.3Ni alloy was reduced by 35% due to hydrogen addition when compressed at 1150 °C/0.01 s$^{-1}$. Under the compression deformation condition of 1200 °C and 0.01 s$^{-1}$, the peak stress of the Ti–45Al–5Nb–0.8Mo–0.3Y alloy was reduced by approximately 25% by hydrogen. Ma et al. [14] studied the effect of hydrogen on the high-temperature deformation behavior of the Ti–44Al–6Nb alloy and found that the alloy containing 0.2 wt.% hydrogen was compressed in a temperature range of 900–1000 °C, and its peak stress was decreased significantly, mainly because hydrogen promoted the dynamic recrystallization and spheroidization of the $\alpha_2$ phase, increased the content of the β phase, and promoted the dynamic recovery of the β phase. Liu et al. [15] found that adding 0.3–0.5 at.% hydrogen in a Ti–47Al alloy could reduce the forging temperature of the alloy from 950 °C to 750 °C without increasing the deformation stress, which was of great significance to reduce mold wear, improve mold service life, and reduce cost.

Previous works have mainly focused on the effect of hydrogen on the mechanical properties and microstructure evolution of TiAl-based alloys under compressive stress. However, some microstructure characteristics and indices on hot workability cannot be observed and investigated under compressive stress, for example, elongation, which is one of the significant indices to reflect the hot workability. Up to now, systematic works on the effect of hydrogen on the hot workability of TiAl-based alloys under tensile stress have not been reported. Therefore, an isothermal tensile test was carried out to investigate the effect of hydrogen on the mechanical properties and microstructure evolution of a TiAl-based alloy in this paper.

## 2. Experiment

The nominal component of the TiAl-based alloy was Ti–45Al–9Nb (at.%). The original experimental material was an as-cast ingot with a diameter of 85 mm and a height of 89 mm. After casting, the ingot was hot isostatic pressed at 1260 °C for 4 h, with a gas pressure of 150 MPa, and then was soaked at 900 °C for 12 h. Finally, it was processed into the tensile specimens by wire cut electrical discharge machining (WEDM) and machining methods. Figure 1 shows the high-temperature tensile test specimen size of the Ti–45Al–9Nb alloy.

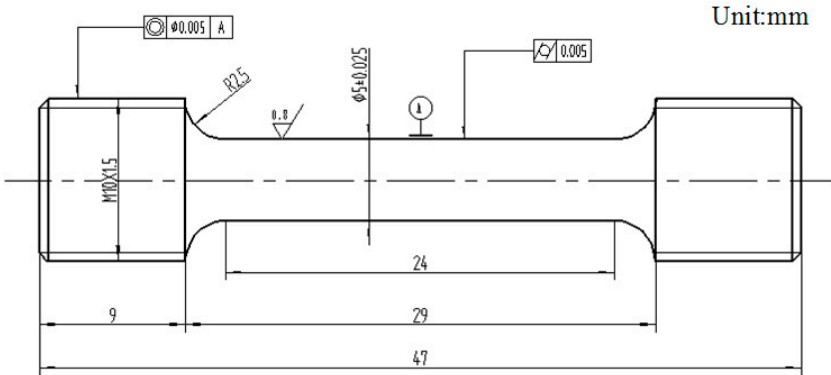

**Figure 1.** High-temperature tensile test specimen size of the Ti–45Al–9Nb alloy.

*2.1. Hydrothermal Treatment*

The tensile samples first underwent high-temperature hydrogenation. The specific process was as follows: first, the samples were placed in acetone for ultrasonic cleaning for 20 min, and then placed in a tube furnace; after vacuum up to $10^{-3}$ Pa, argon gas was filled; after the furnace temperature rose to 800 °C, the hydrogen was filled with an absolute pressure of 0.1–0.15 MPa, and then the samples were soaked for 2 h. When the furnace was cooled to room temperature, samples with H content of 0.8 at.% (abbreviated as 0.8 H below) were finally obtained. The highest hydrogen content obtained by the current hydrogenation equipment was only 0.8 at.%, which had the most significant effect on the mechanical properties and microstructure evolution of the present alloy, and so the study focused on a 0.8 at.% hydrogen. The hydrogen content was examined by a LECO-ROH600 oxygen/hydrogen analyzer (LECO, St Joseph, MI, USA), with an accuracy of 0.01 ppm. The error of the hydrogen content was ±3%. In order to accurately compare and study the effect of H on the high-temperature tensile deformation behavior and microstructure evolution of the Ti–45Al–9Nb alloy, the samples without hydrogen had a vacuum heat treatment with the same heat treatment system (without hydrogen addition).

*2.2. High-Temperature Tensile Test*

High-temperature uniaxial tensile test of the unhydrogenated and hydrogenated Ti–45Al–9Nb alloy samples was carried out on an MTS 880 universal tensile test machine by using the equivalent strain rate tensile method. The specific experimental process was as follows: first, the surface of the samples was sprayed with antioxidant alumina to prevent the surface oxidation; second, the samples were heated to the test temperature in a three-section circular resistance furnace, and had heat preservation for 10 min. The test temperatures were 1050, 1100, and 1150 °C and the strain rates were 0.0004, 0.001, and 0.0025 s$^{-1}$, respectively, and water-quenching was carried out immediately after the test. Finally, the stress–strain curves and deformed samples were obtained. The deformation behaviors of the alloy at the temperature of 1150 °C and the strain rate of 0.0004–0.0025 s$^{-1}$ were mainly investigated in this paper.

*2.3. Microstructural Analysis*

The gauge part of samples after tensile deformation was wire-electrode cut, and then the surface to be observed was ground with 240, 400, and 600-grit SiC papers. Finally, electropolishing was carried out. The electrolytic polishing solution was 60% methanol + 34% n-butanol + 6% perchloric acid, the power supply voltage was adjusted to 20 V, the current maintained at 0.5–0.6 A, and the electrolysis time was 50 s.

Scanning electron microscopy (SEM) (JSM-7800F, Jeol, Tokyo, Japan) was used to analyze the microstructure of the gauge part of samples after high-temperature tension.

An X-ray diffractometer (XRD) (BRUKER Company, Karlsruhe, Germany, model: D8 ADVANCE) was used to analyze the phase. The radiation light used in the experiment was Cu K$\alpha$ with a wavelength of 1.5418 Å, the generator's power was 1.6 kW (40 kV, 40 mA), the continuous scanning range was 10°–90°, the scanning rate was 0.2°/s, the step scanning step length was 0.02°, and each step lasted for one second.

## 3. Results and Discussion

### 3.1. Effect of Hydrogen on the Microstructure of Alloy at Room Temperature

SEM microstructures of the unhydrogenated and hydrogenated alloys are shown in Figure 2. Note that the dark gray phase is the $\gamma$ phase, the light gray phase is the $\alpha_2$ phase, and the bright white phase is the B2 phase. The $\alpha_2$ phase is the ordered phase of the $\alpha$ phase at low temperature, and the B2 phase is the ordered phase of the $\beta$ phase at low temperature. The microstructures of both alloys were near-lamellar, which were mainly composed of $\gamma/\alpha_2$-lamellar colonies with an average size of about 800 μm. In addition, a small number of equiaxed $\gamma$ grains and irregular B2 grains were distributed along the lamellar boundaries.

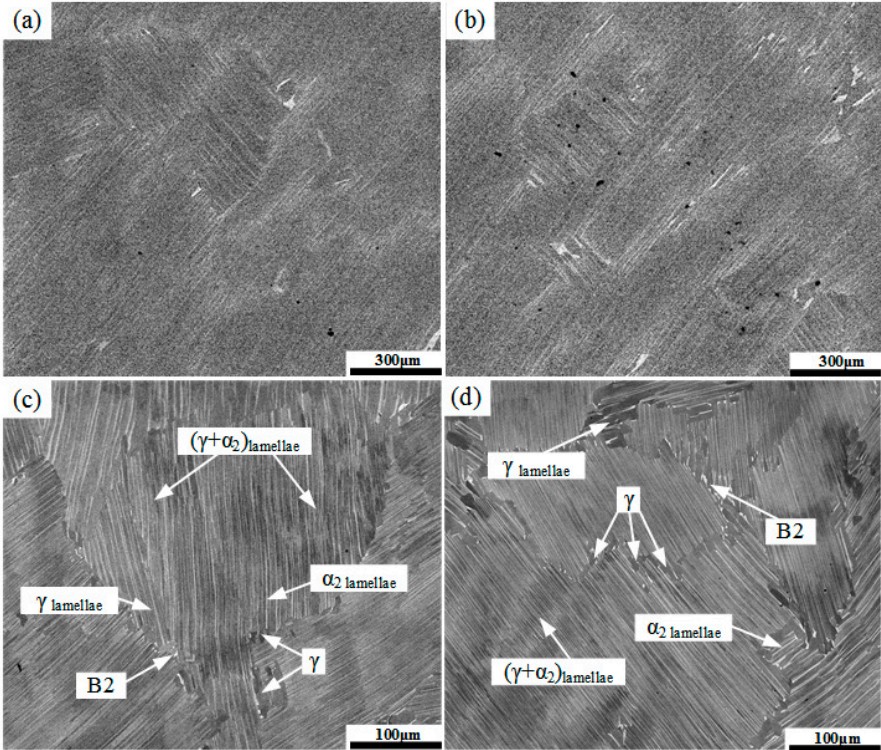

**Figure 2.** SEM microstructures of the unhydrogenated (**a**,**c**) and hydrogenated (**b**,**d**) Ti–45Al–9Nb alloys. (**c**,**d**) are the magnified images of (**a**,**b**), respectively.

Figure 3 shows the X-ray diffraction patterns of the unhydrogenated and hydrogenated Ti–45Al–9Nb alloys. Both the unhydrogenated and hydrogenated alloys were composed of a large amount of the $\gamma$ phase (L1$_0$ crystal structure, $a = b = 0.4005$ nm, $c = 0.407$ nm, $a/c = 0.984$), a certain amount of the $\alpha_2$ phase (D0$_{19}$ crystal structure, $a = b = 0.578$ nm, $c = 0.465$ nm, $a/c = 1.243$), and a very small amount of the B2 phase (CsC1 crystal structure, $a = b = c = 0.316$ nm). The diffraction peaks of the $\alpha_2$ phase, $\gamma$ phase, and B2 phase were basically unchanged after hydrogen addition, and the intensity of the diffraction peaks of the $\alpha_2$ phase and B2 phase was slightly stronger than that of the unhydrogenated alloy, indicating that hydrogen increased the content of the $\alpha_2$ phase and B2 phase. In addition, the diffraction peak of the (TiAl)H$_x$ hydride was found at $2\theta = 35.46°$ after hydrogen

addition. Meanwhile, the hydride had a tetragonal crystal structure with lattice constants *a* = 0.452 nm, *c* = 0.326 nm, and *c/a* = 0.721 [16]. The hydrogenation treatment of TiAl-based alloys was achieved by the diffusion of hydrogen atoms. In the diffusion process, hydrogen was first decomposed into hydrogen atoms and bumped into the surface of the samples. Due to a large number of defects and higher energy in the grain boundary or phase boundary, a channel was provided for the diffusion of hydrogen atoms. Therefore, hydrogen atoms preferentially diffused in a short range along the grain boundary or phase boundary, so the hydrogen concentration at the grain boundary or phase boundary reached saturation in a short time. Therefore, the concentration of hydrogen atoms at the grain boundary or phase boundary was relatively high, which could easily meet the requirements of composition fluctuation and energy fluctuation for hydride nucleation. When the hydrogen content exceeded its saturated solid solubility, the hydrogen combined with titanium aluminum to form titanium aluminum hydride.

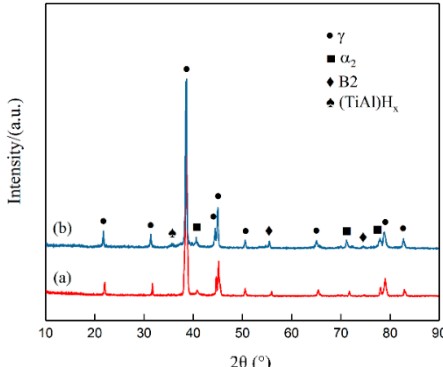

**Figure 3.** XRD patterns of the unhydrogenated (**a**) and hydrogenated (**b**) Ti–45Al–9Nb alloys.

In order to quantitatively study the content of each phase in the unhydrogenated and hydrogenated alloys, a quantitative XRD analysis was conducted. Figure 4 shows the relative volume fraction of the $\gamma$, $\alpha_2$, B2, and (TiAl)H$_x$ phases in the unhydrogenated and hydrogenated alloys, which are calculated based on the integration area of the diffraction peaks. The contents of the $\gamma$, $\alpha_2$, and B2 phases in the unhydrogenated alloy were 77.54%, 20.89%, and 1.57%, respectively, while the contents of the $\gamma$, $\alpha_2$, and B2 phase in the hydrogenated alloy were 69.31%, 26.18%, and 2.85%, respectively. In general, hydrogen treatment can reduce the content of the $\gamma$ phase because adding H can effectively promote the diffusion of elements and distort the $\gamma$ phase lattice, thus promoting the $\gamma \rightarrow \alpha_2$ phase transformation [17]. In addition, the content of the B2 phase in the hydrogenated alloy was also slightly increased, indicating that hydrogen can stabilize the B2 phase and promote its precipitation.

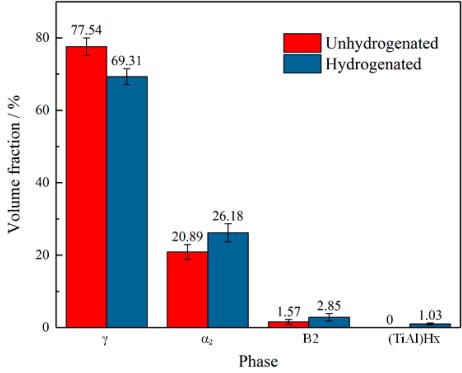

**Figure 4.** Volume fraction of phases in the unhydrogenated and hydrogenated Ti–45Al–9Nb alloys.

*3.2. High-Temperature Flow Behavior of Hydrogenated Alloy*

3.2.1. True Stress–True Strain Curves and Their Characteristics

Figure 5 shows the tensile deformed specimens and true stress–true strain curves of the unhydrogenated and hydrogenated Ti–45Al–9Nb alloys. Figure 5a shows the deformed specimens. Deformed at 1150 °C, all samples underwent a certain plastic deformation. Under the same deformation condition, the plastic deformation degree of the unhydrogenated alloy was greater than that of the hydrogenated alloy. Figure 5b–d show the true stress–true strain curves of the unhydrogenated and hydrogenated Ti–45Al–9Nb alloy samples deformed at 1150 °C, with strain rates of 0.0004, 0.001 and 0.0025 s$^{-1}$, respectively. Works have reported that if the stress dramatically drops after peak stress with increasing strain, the stress–strain curve is related to dynamic recrystallization [18]. Accordingly, the stress–strain curves shown in Figure 5b–d are supposed to be related to dynamic recrystallization. In the early stage of deformation, dislocation movement was gradually obstructed by increasing the dislocation propagation, resulting in a dislocation pileup, which increased the dislocation density and formed dislocation tangles. Meanwhile, a great stress concentration would be generated at the junction of lamellar colonies. This dislocation tangle and stress concentration would increase the flow stress and lead to work hardening, and macroscopically, the stress increased rapidly with the increase in strain until the stress reached its peak [19]. Subsequently, the stress decreased with an increase in the strain, which was mainly attributed to the increase of dynamic recrystallization. Dynamic recrystallization softening and the work hardening phase offset each other. When the softening effect of dynamic recrystallization was greater than the hardening effect of hot working, the strain tended to decrease significantly with the increase of the strain.

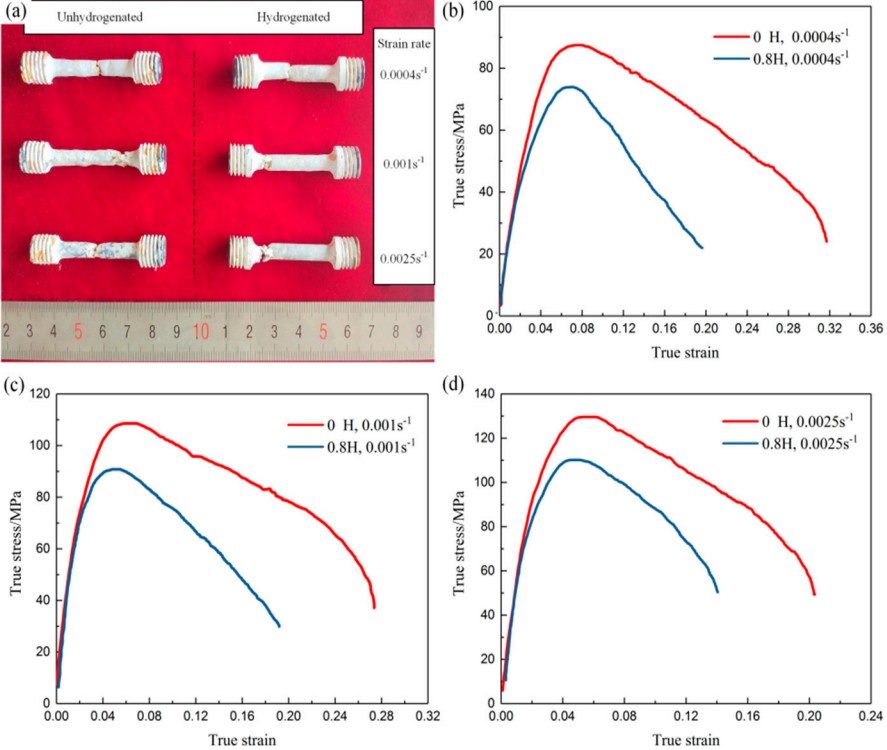

**Figure 5.** Tensile deformed specimens (**a**) and true stress-true strain curves (**b–d**) of the unhydrogenated and hydrogenated Ti–45Al–9Nb alloys deformed at (**b**) 1150 °C/0.0004 s$^{-1}$, (**c**) 1150 °C/0.001 s$^{-1}$, and (**d**) 1150 °C/0.0025 s$^{-1}$.

In addition, under the same deformation condition, the stress level of the hydrogenated alloy was lower than that of the unhydrogenated alloy, and the peak strain (the strain corresponded by the peak

stress) of the hydrogenated alloy was lower than that of the unhydrogenated alloy. The smaller the peak strain, the sooner the dynamic recrystallization occurred [20]. The effects of hydrogen on the peak stress and elongation of the Ti–45Al–9Nb alloy at different strain rates are discussed in detail below.

### 3.2.2. Effect of Strain Rate on Flow Stress and Elongation

Figure 6 shows the peak stress and elongation of the unhydrogenated and hydrogenated Ti–45Al–9Nb alloys deformed in a strain rate range 0.0004–0.0025 s$^{-1}$, with a temperature of 1150 °C. As can be seen from Figure 6a,b, the peak stresses of the hydrogenated alloy and hydrogen decreased with the decrease in the strain rate, and the decrease trend tended to become flatter with the decrease in strain rate. At the same strain rate, the stress level of the hydrogenated alloy was lower than that of the unhydrogenated alloy, and the decrease rate of the peak stress increased with the decrease in the strain rate. Obviously, the addition of hydrogen caused flow softening, and the effect of hydrogen-induced softening was more obvious when the strain rate was lower. When deformed at 0.0004 s$^{-1}$, the decrease rate of the peak stress was more obvious, which was (16.28 ± 0.17)% lower than that of the unhydrogenated alloy. The softening mechanism of the hydrogenated alloy mainly includes dynamic recovery and dynamic recrystallization. In the plastic deformation of TiAl-based alloys, dislocation slip and climbing usually occur [21]. When the deformation temperature is constant, dislocation glide and climb gain more time with the decrease in strain rate, which allows the dynamic recrystallization nucleation to take place more easily to some extent. Therefore, the peak stress of both the hydrogenated and hydrogenated alloys decreased with the decrease in strain rate.

In addition, from the perspective of dislocation velocity and critical shear stress, the increase in strain rate would increase the dislocation movement and further increase the critical shear stress of the dislocation movement. Their relationship can be expressed as follows [22]:

$$v = v_0 e^{\left(\frac{C}{T\tau}\right)} \tag{1}$$

where $v$ is the velocity dislocation movement; $v_0$ is the sound's propagation speed in titanium aluminum alloy; $C$ is the material constant; $T$ is the absolute temperature; and $\tau$ is the critical shear stress of dislocation movement.

According to Equation (1), under the condition of constant deformation temperature, the increase of $v$ inevitably leads to the increase of $\tau$, that is, the flow stress increases.

Figure 6c,d show the reduction rate of the elongation of the hydrogenated alloy when the temperature was 1150 °C and the strain rate was 0.0004–0.0025 s$^{-1}$. As can be seen from Figure 6c,d, the elongation of the alloy decreased with the increase in the strain rate. At the same deformation temperature, the elongation of the hydrogenated alloy was lower than that of the unhydrogenated alloy, which decreased by (26.05 ± 0.45)% (0.0004 s$^{-1}$), (23.49 ± 0.38)% (0.001 s$^{-1}$), and (14.23 ± 0.19)% (0.0025 s$^{-1}$), respectively, indicating that 0.8 at.% H reduced the high-temperature plasticity of Ti–45Al–9Nb under this deformation condition. This might be due to the fact that, in the hydrogenated alloy, hydrogen did not always exist in the alloy in the form of a solid solution, and that some hydrogen combined with alloy atoms to form hydride (TiAl)H$_x$. The results indicated that the hydride itself was a brittle phase, and that it could easily become a crack source and promote the generation of cracks [23,24]. Therefore, the plasticity of the hydrogenated alloy was reduced due to the existence of hydride (TiAl)H$_x$. In addition, as shown in Figure 6d, the reduction rate of hydrogenated elongation became smaller and smaller with the increase in the strain rate. This was mainly because with the increase in strain rate, the deformation time of the unhydrogenated alloy became shorter and shorter, and the dynamic recrystallization grain nucleation and growth were less likely to take place. Meanwhile, when the layer of lamellar colonies inside the plastic deformation was small, the pressure under the action of relative rotation took place between lamellar colonies, which easily caused the stress concentration in the process of the rotation of the lamellar colonies, resulting in too early deformation and instability of

the alloy. Therefore, the hydrogenated alloy showed a significant reduction trend of elongation in the macroscopic view.

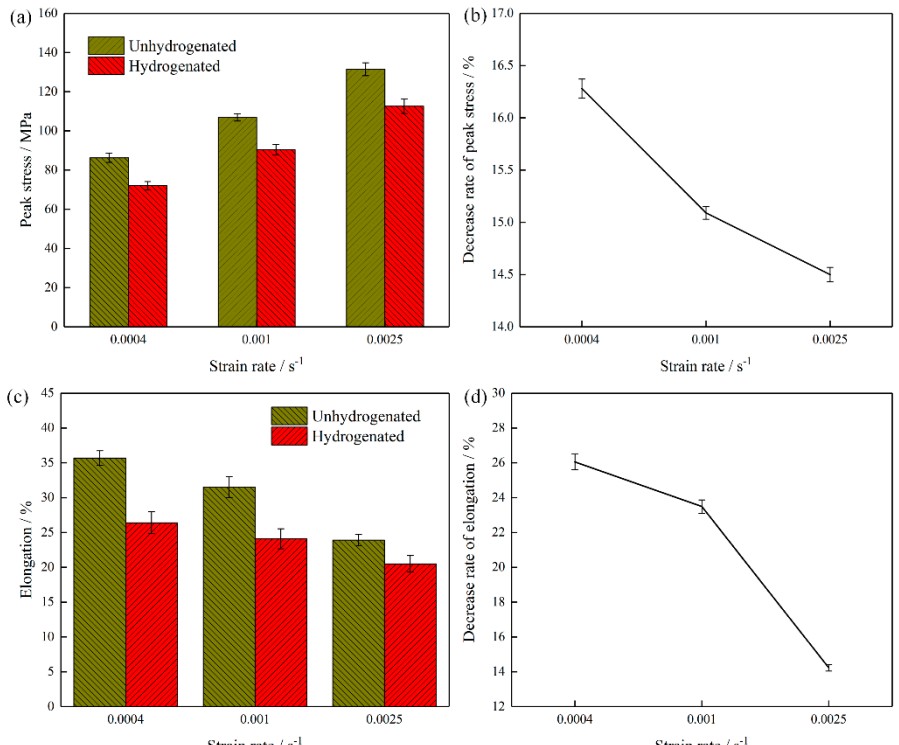

**Figure 6.** Peak stress and elongation of the unhydrogenated and hydrogenated Ti–45Al–9Nb alloys deformed in a strain rate range of 0.0004–0.0025 s$^{-1}$, with a temperature of 1150 °C. (**a**) Peak stresses and (**b**) the decrease rate of peak stress due to hydrogen addition, (**c**) elongation, and (**d**) the decrease rate of elongation due to hydrogen addition.

### 3.2.3. Constitutive Equation at High Temperature

A large number of works have indicated that the high-temperature deformation process of metal materials such as steel and iron materials, aluminum alloy, titanium alloy and TiAl-based alloy is through the thermal activation process. High-temperature flow behavior is controlled by deformation temperature and strain rate, and there is a constitutive relationship between flow stress σ, deformation temperature T, and strain rate $\dot{\varepsilon}$ (i.e., hyperbolic sine function), as shown in Equation (2) [25,26]:

$$\dot{\varepsilon} = A[\sin h(\alpha\sigma)]^n \exp\left(-\frac{Q}{RT}\right) \tag{2}$$

The relationship among σ, $\dot{\varepsilon}$, and *T* at a low stress (ασ < 0.8) is expressed as an exponential function:

$$\dot{\varepsilon} = A_1\sigma^n \exp\left(-\frac{Q}{RT}\right) \tag{3}$$

The relationship among σ, $\dot{\varepsilon}$, and *T* at a high stress (ασ > 0.8) is expressed as an exponential function:

$$\dot{\varepsilon} = A_2[\exp(\beta\sigma)]\exp\left(-\frac{Q}{RT}\right) \tag{4}$$

where *n* and $n_1$ are stress exponents, $A$, $A_1$, $A_2$; α and β are the material constants, among which α, β, and $n_1$ satisfies the relationship α = β/$n_1$; *R* is the gas constant; *Q* is the thermal deformation activation energy (KJ/mol); and *T* is the absolute temperature (K).

Assuming that deformation activation energy $Q$ is independent of the deformation temperature $T$, and natural logarithms of both sides of the above three equations can be obtained as follows:

$$\ln \dot{\varepsilon} + \frac{Q}{RT} = \ln A + n\ln[\sin h(\alpha\sigma)] \tag{5}$$

$$\ln \dot{\varepsilon} + \frac{Q}{RT} = \ln A_1 + n_1 \ln \sigma \tag{6}$$

$$\ln \dot{\varepsilon} + \frac{Q}{RT} = \ln A_2 + \beta\sigma \tag{7}$$

The partial derivatives of both sides of Equations (2)–(4) were obtained, and then $n_1$, $\beta$, and $n$ can be expressed as follows:

$$n = \left. \frac{\partial(\ln \dot{\varepsilon})}{\partial\{\ln[\sin h(\alpha\sigma)]\}} \right|_{1/T} \tag{8}$$

$$n_1 = \left. \frac{\partial(\ln \dot{\varepsilon})}{\partial(\ln \sigma)} \right|_{1/T} \tag{9}$$

$$\beta = \left. \frac{\partial(\ln \dot{\varepsilon})}{\partial\sigma} \right|_{1/T} \tag{10}$$

The partial derivatives of both sides of Equations (5)–(7) were obtained, and then substituted into Equations (8)–(10) to calculate the expressions of $n_1$, $\beta$, and $n$, and the activation energy $Q$ can be calculated.

All stress conditions:

$$Q = R\left. \frac{\partial(\ln \dot{\varepsilon})}{\partial\{\ln[\sin h(\alpha\sigma)]\}} \right|_{1/T} \cdot \left\{ \frac{\partial\{\ln[\sin h(\alpha\sigma)]\}}{\partial(1/T)} \right\}\Bigg|_{\dot{\varepsilon}} \tag{11}$$

Low stress:

$$Q = R\left. \frac{\partial(\ln \dot{\varepsilon})}{\partial(\ln \sigma)} \right|_{1/T} \cdot \left. \frac{\partial(\ln \sigma)}{\partial[\ln(1/T)]} \right|_{\dot{\varepsilon}} \tag{12}$$

High stress:

$$Q = R\left. \frac{\partial(\ln \dot{\varepsilon})}{\partial(\ln \sigma)} \right|_{1/T} \cdot \left\{ \frac{\partial\sigma}{\partial[\ln(1/T)]} \right\}\Bigg|_{\dot{\varepsilon}} \tag{13}$$

The hyperbolic sine function (see Equation (2)) is more suitable to express the relationship between the peak stress and strain rate of the unhydrogenated and hydrogenated TiAl-based alloys [27]. According to the above-mentioned equations and experimental data, the relationships between the peak stress and strain rate of the unhydrogenated and hydrogenated Ti–45Al–9Nb alloys can be built. Figure 7 shows the relationships between $\sigma_p$ and $\dot{\varepsilon}$ of the unhydrogenated and hydrogenated alloys. According to the curve fitting results, unary linear regression was carried out, the constant $n_1$ and $\beta$ of the hydrogenated and unhydrogenated alloys could be obtained, where their $n_1$ were 4.55 ± 0.17 and 4.17 ± 0.13, respectively, and their $\beta$ were 0.04 ± 0.02 and 0.05 ± 0.02, respectively. Later, the value of the $\alpha$ of the hydrogenated and unhydrogenated alloys can be calculated by the equation $\alpha = \beta/n_1$ as 0.009 ± 0.001 and 0.011 ± 0.001, respectively.

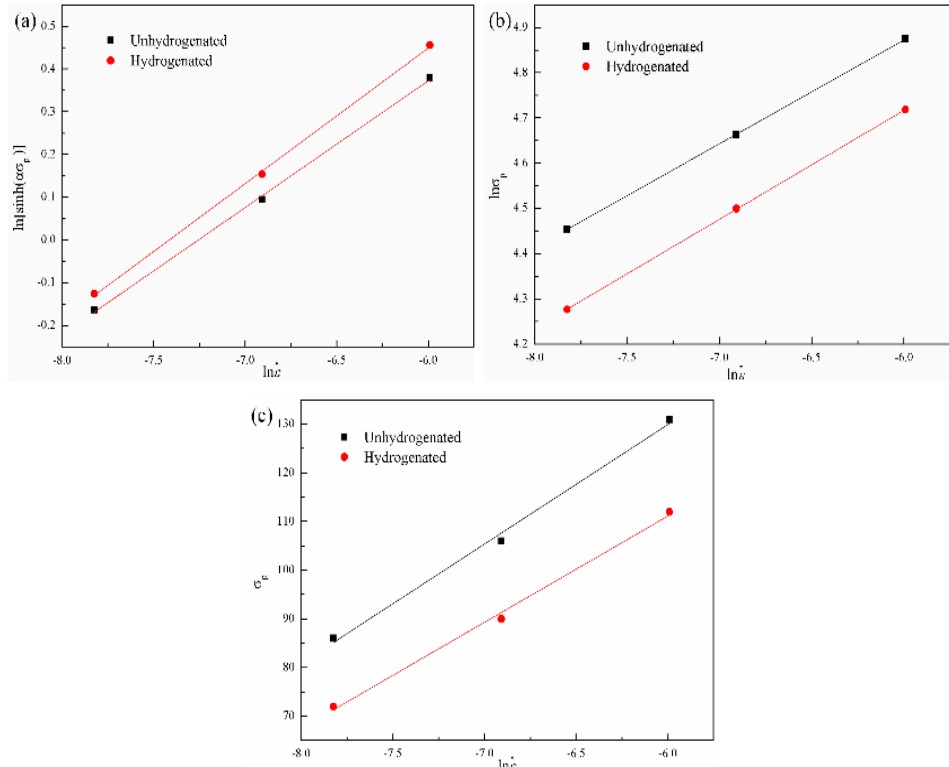

**Figure 7.** Relationships between $\sigma_p$ and $\dot{\varepsilon}$ of the unhydrogenated and hydrogenated alloys: (**a**) $\ln[\sinh(\alpha\sigma_p)]$–$\ln\dot{\varepsilon}$; (**b**) $\ln\sigma_p$–$\ln\dot{\varepsilon}$ and (**c**) $\sigma_p$–$\ln\dot{\varepsilon}$ relationships.

$\alpha$ can be used to obtain the relation curve of $\ln\dot{\varepsilon}$–$\ln[\sinh(\alpha\sigma_p)]$, as shown in Figure 7c. According to Equation (8), the value of $n$ can be calculated. The $n$ values of the hydrogenated and unhydrogenated alloys were $3.37 \pm 0.12$ and $3.15 \pm 0.11$, respectively. Finally, the deformation activation energy $Q$ can be calculated from the relationship between temperature $T$ and strain rate $\dot{\varepsilon}$ of the unhydrogenated and hydrogenated alloys, as shown in Table 1. The $Q$ values of the hydrogenated and unhydrogenated alloys were $(584.31 \pm 5.34)$ KJ/mol and $(556.95 \pm 4.15)$ KJ/mol, respectively.

**Table 1.** Constitutive equation parameter values of the hydrogenated and unhydrogenated alloys.

| Samples | ln*A* | *n* | $\alpha$ | *Q* (KJ/mol) |
|---|---|---|---|---|
| Unhydrogenated | 42.31 ± 0.83 | 3.37 ± 0.12 | 0.009 ± 0.001 | 584.31 ± 5.34 |
| Hydrogenated | 39.97 ± 0.69 | 3.15 ± 0.11 | 0.011 ± 0.001 | 556.95 ± 4.15 |

The constitutive equations of the Ti–45Al–9Nb alloy deformed at high-temperature can be obtained as follows:

Unhydrogenated:

$$\dot{\varepsilon} = e^{42.31}[\sin h(0.009\sigma)]^{3.37}\exp\left(-\frac{584310}{RT}\right) \tag{14}$$

Hydrogenated:

$$\dot{\varepsilon} = e^{39.97}[\sin h(0.011\,\sigma)]^{3.15}\exp\left(-\frac{556950}{RT}\right) \tag{15}$$

The above analysis indicates that for the Ti–45Al–9Nb alloy, the stress and deformation condition satisfied a hyperbolic sinusoidal relationship at high-temperature, and the deformation activation energy was significantly higher than the energy required for self-diffusion (290–345 KJ/mol) [28]. Therefore, the main softening mechanism for the unhydrogenated and hydrogenated alloys was dynamic recrystallization. Hydrogen decreased the deformation activation energy, which was mainly

attributed to the fact that hydrogen could reduce the diffusion barrier, increase the diffusion coefficient of atoms, and coordinate deformation during thermal deformation. At the same time, hydrogen could promote dislocation movement, which led to a decrease in the thermal deformation activation energy.

*3.3. Microstructure Evolution*

　　Figure 8 shows the SEM images of the unhydrogenated and hydrogenated alloys deformed at 1150 °C and in the strain rate range of 0.0004–0025 s$^{-1}$. With the decrease in the strain rate, the deformation and the extent of dynamic recrystallization increased gradually. This is because the alloying element diffused sufficiently at a lower strain rate, which facilitated the decomposition of $\alpha_2$ and $\gamma$ phase lamellae. Therefore, the dynamic recrystallization took place more sufficiently. The decomposition of lamellae can also be regarded as a special dynamic recrystallization (DRX) process. It also includes nucleation (lamellar fragmentation) and growth (lamellar spheroidization) [29]. When the strain rate was 0.0025 s$^{-1}$, some lamellar colonies of the unhydrogenated and hydrogenated alloys were bent to a certain extent, showing a wavy shape. The orientation of the bent and deformed $\alpha_2/\gamma$ lamellar was a medium orientation, that is, the lamellar direction was perpendicular to the tensile direction. A small amount of block-shaped $\alpha_2$ phase was formed at the grain boundary of the lamellar colonies in the unhydrogenated alloy. However, the hydrogenated alloy had fractured when the deformation was very small, thus it was difficult to observe the fracture or dynamic recrystallization structure. When the strain rate decreased to 0.001 s$^{-1}$, there were not only bending lamellar, but also a certain amount of lamellar fragmentation and spheroidized structures as well as fine dynamic recrystallization grains in the unhydrogenated alloy. It can be seen from Figure 8b,e that some lamellar colonies in the unhydrogenated alloy increased in spacing along the direction of stress, that is, the lamellar was obviously coarsened and accompanied by lamellar bending. The $\gamma$ phase recrystallization grains occurred in the $\alpha_2/\gamma$ lamellar interface. As the number of $\alpha_2$-phase slip systems was less than that of the $\gamma$ phase, uncoordinated deformation and stress concentration easily occurred in the lamellar interface, which caused the lamellar interface to have local small deformation and provided driving energy for the recrystallization nucleation [30]. However, only a few lamellae were coarsened and the recrystallization grains were relatively less. When the strain rate dropped to 0.0004 s$^{-1}$, there were many bent and elongated lamellae. The spacing of the lamellar colonies was larger, the fragmentation of the lamellar colonies was enhanced, and the number and size of the recrystallization grains at the interface of the lamellar colonies increased obviously. In comparison, there were more residual lamellae and the extent of recrystallization was lower than that of the unhydrogenated alloy. According to the above analysis, dynamic recrystallization was very sensitive to the strain rate, and the smaller the strain rate, the more obvious the dynamic recrystallization. Under the same deformation condition, the deformation extent of the hydrogenated alloy was less than that of the unhydrogenated alloy, which was consistent with the elongation decrease in the hydrogenated alloy above-mentioned.

　　Through a comparison with previous work, it was found that the influence of hydrogen on the mechanical properties of the alloy was different during the high-temperature plastic deformation of the hydrogenated TiAl alloys. When the specimen was subject to tensile stress, hydrogen deteriorated the elongation of the Ti–45Al–9Nb alloy at high temperature. The main reason is that there are two forms of hydrogen in the alloys. On one hand, hydrogen atoms are dissolved in the lattice interstices. On the other hand, hydrogen atoms combine with alloy atoms to form hydride (i.e., (TiAl)H$_x$). After hydrogen entered the lattice sites, it weakens the binding force between the Ti and Al atoms, and reduces the binding energy [31]. The aggregation of hydrogen atoms along the grain boundary or phase boundary decreases the driving force for dislocation emission and movement, which leads to local plastic deformation and reduces the toughness. Meanwhile, the hydride gathers along the grain boundary, which is a source of cracking. The tensile stress accelerates the emergence and propagation of the grain boundary cracks, and finally results in deformation instability. Therefore, the elongation of the hydrogenated alloy was less than that of the unhydrogenated alloy. For most hydrogenated TiAl alloys using the solid hydrogenation technique, there is no hydride when compressive tests are conducted at

high temperature [13–15]. When such hydrogenated specimens are compressed at high temperature, hydrogen can improve the plastic deformability, which is mainly due to hydrogen-induced dislocation movement, hydrogen-promoted dynamic recrystallization and twinning, and hydrogen-increased β phase content.

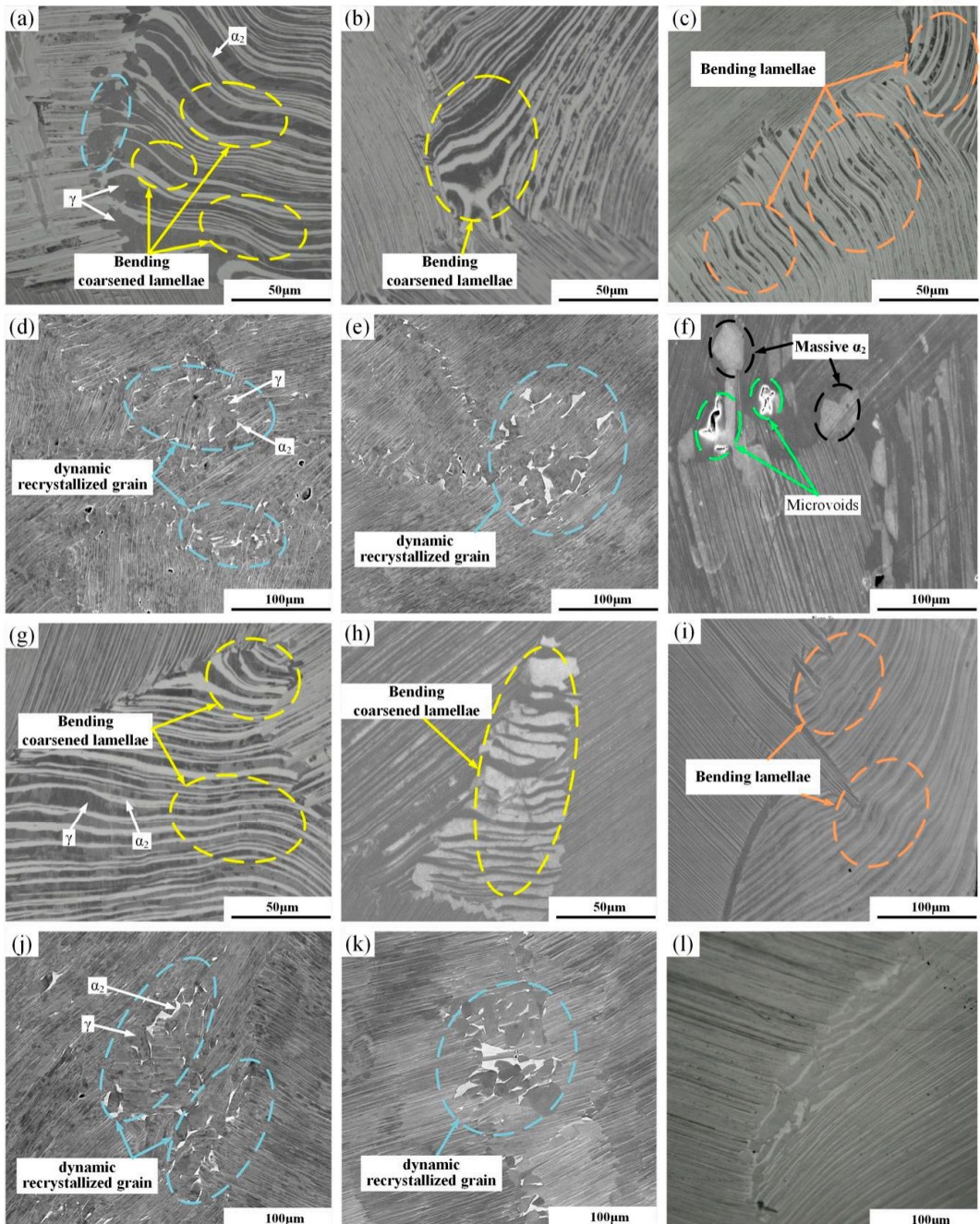

**Figure 8.** SEM images of the unhydrogenated (**a–f**) and hydrogenated (**g–l**) alloys deformed at (**a,d,g,j**) 1150 °C/0.0004 s$^{-1}$, (**b,e,h,k**) 1150 °C/0.001 s$^{-1}$, and (**c,f,i,l**) 1150 °C/0.0025 s$^{-1}$.

In order to study the mechanism of crack generation and propagation in hydrogen-containing alloys, we observed the cracks generated by the alloy under different deformation conditions and found that the internal crack propagation mechanism of the alloy was similar under this test condition. Therefore, crack generation and propagation under the deformation condition of 1150 °C/0.001 s$^{-1}$ was chosen as the main research object. Figure 9 shows the cracking modes of the hydrogenated

alloy deformed at 1150 °C/0.001 s$^{-1}$. The cracks mainly occurred in the internal lamellar colonies or at the boundary of the lamellar colonies. Among them, the internal cracks in the lamellar colonies were mainly generated at the $\alpha_2/\gamma$ lamellar boundary, which could be divided into inter-lamellar and trans-lamellar cracks, according to the different propagation techniques of cracks. These two kinds of cracks were mostly wedge cracks, which were mainly generated at the $\alpha_2/\gamma$ lamellar boundary and were caused by the deformation disharmony between the $\alpha_2$ and $\gamma$ lamellae. After crack nucleation was completed along the flat $\alpha_2/\gamma$ lamellar boundary, the crack continued to propagate along the $\alpha_2/\gamma$ lamellar boundary until reaching the lamellar colony boundaries, as shown in Figure 9a. Trans-lamellar cracks nucleated in the crooked $\alpha_2/\gamma$ lamellar boundary, and the crooked $\alpha_2/\gamma$ phase boundary retarded the propagation of cracks to some extent. These types of cracks were usually accompanied by bridging structures, as shown in Figure 9b. The cracks along the lamellar colony boundary were mainly generated in the $\alpha_2/\gamma$ lamellar colony boundary or the $\gamma/\gamma$ lamellar colony boundary. Such cracks in the hydrogenated alloy were more than that in the unhydrogenated alloy. The accumulation of solid dissolved hydrogen and precipitated hydride at the grain boundary caused the stress concentration at the grain boundary, reduced the binding force at the boundary, and weakened lamellar colony boundaries, thus leading to more along-lamellar colony boundary cracks [32], as shown in Figure 9c. The propagation of such cracks was different from that of the inter-lamellar cracks. The along-lamellar colony boundary cracks formed a cavity after completing nucleation. Due to the occurrence of deformation in the alloy, crystal defects such as vacancies continuously increased, and in order to reduce the surface energy, these vacancies gathered along a certain direction under external stress, which made the cavity gradually grow and form a series of "cavity beads" along the lamellar colony boundaries. The cavities mutually combined to complete the crack propagation. The crack shown in Figure 9d is a combination of the above three types of cracks. Figure 10 is a schematic diagram of inter-lamellar, trans-lamellar, and along-lamellar colony boundary cracking propagation of the hydrogenated alloy.

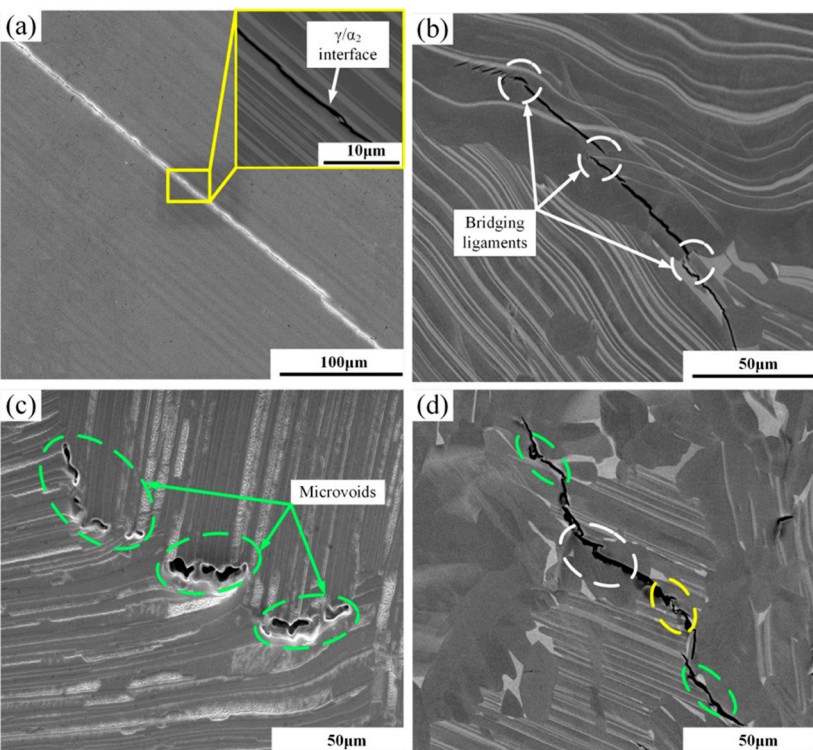

**Figure 9.** Cracking modes of the hydrogenated alloy deformed at 1150 °C/0.001 s$^{-1}$: (**a**) inter-lamellar crack, (**b**) trans-lamellar crack, (**c**) along-lamellar colony boundary crack, and (**d**) mixed mode crack.

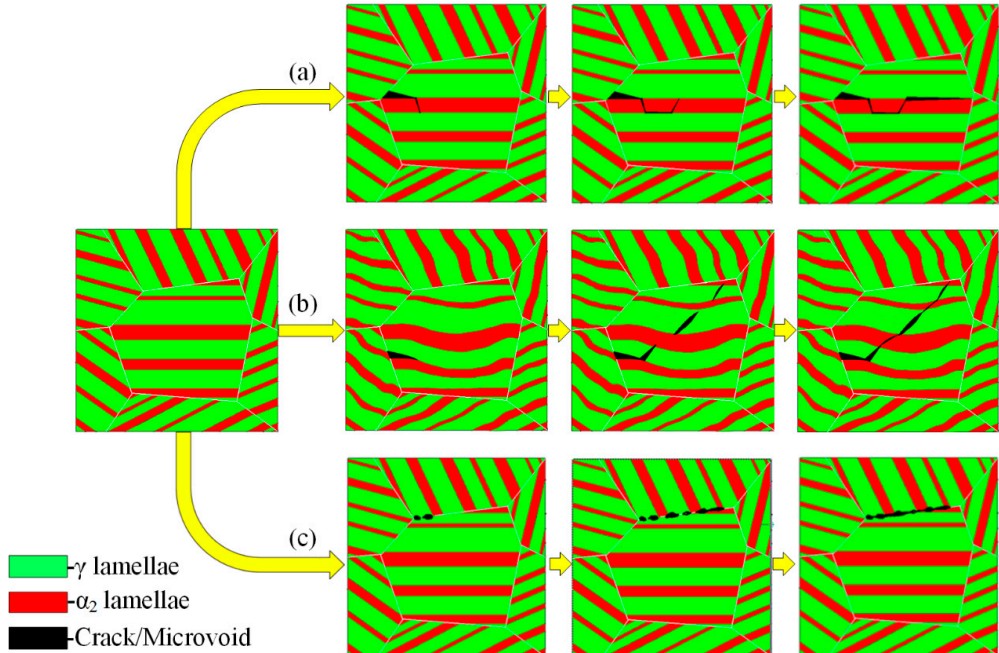

**Figure 10.** Schematic diagram of inter-lamellar (**a**), trans-lamellar (**b**), and along-lamellar colony boundary (**c**) cracking propagation of the hydrogenated alloy.

## 4. Conclusions

- Through hydrogen treatment, the phase content of the $\alpha_2$ and B2 phases in the hydrogenated alloy increased when compared with that of the unhydrogenated alloy. In the hydrogenated alloy, hydride $(TiAl)H_x$ was observed, which led to cracks.

- Compared with the unhydrogenated alloy, the flow stress of the hydrogen alloy was significantly reduced (i.e., hydrogen-induced softening). Hydrogen-induced softening was enhanced with a decrease in the strain rate. Deformed at 1150 °C/0.0004 s$^{-1}$, the peak stress was decreased by $(16.28 \pm 0.17)\%$ due to hydrogen addition. The elongation of the hydrogenated alloys was decreased by $(26.05 \pm 0.45)\%$ $(0.0004\,s^{-1})$, $(23.49 \pm 0.38)\%$ $(0.001\,s^{-1})$, and $(14.23 \pm 0.19)\%$ $(0.0025\,s^{-1})$, indicating that the addition of 0.8 at.% H reduced the high-temperature plasticity of Ti–45Al–9Nb alloy. In addition, the deformation activation energy of the hydrogenated alloy was lower than that of the unhydrogenated alloy.

- Under the same deformation condition, the deformation extent of the hydrogenated alloy was less than that of the unhydrogenated alloy. Accordingly, more residual lamellae and the lower extent of recrystallization were observed in the hydrogenated alloy. In addition, there were three types of cracks in the hydrogenated alloy (i.e., inter-lamellar, trans-lamellar, and along-lamellar colony boundary cracks). Furthermore, more along-lamellar colony boundary cracks occurred in the hydrogenated alloy.

**Author Contributions:** S.W. (Shouren Wang) and D.W. developed the experimental plan. Q.Y. wrote the main part of the manuscript. Q.Y. and S.W. (Shuxu Wu) tested the high temperature tensile properties of the different alloys and established the constitutive equation. Q.Y. and B.K. performed the characterization of the microstructure of the samples. Q.Y. and T.X. conducted the final typesetting of the article. S.W. (Shouren Wang) and D.W. summed up the article and conducted a final review. All authors have read and agreed to the published version of the manuscript.

**Funding:** This research was funded by the National Natural Science Foundation of China (51705199 and 51872122), the Natural Science Foundation of Shandong Province (ZR2017BEE055), the Key Technology Research and Development Program of Shandong (2019GGX104045 and 2017GGX30143), the Program for PhD Research Start-up Fund of Shandong Jiaotong University (Z201709), and the Taishan Scholar Engineering Special Funding (ts201511040).

**Acknowledgments:** We sincerely appreciate the support on the microstructure observation and analysis from the Analysis and Testing Center of Shandong University.

**Conflicts of Interest:** The authors declare no conflict of interest.

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
