# Peer review of "Effect of 0.8 at.% H on the Mechanical Properties and Microstructure Evolution of a Ti–45Al–9Nb Alloy Under Uniaxial Tension at High Temperature"

_coatings, doi:10.3390/coatings10010052_

Round 1

Reviewer 1 Report

The manuscript describes the effect of 0.8 at.%H on mechanical properties and microstructure evolution of a Ti-45Al-9Nb alloy under uniaxial tension at high temperature. The authors show that the stress level and peak strain of the Ti-45Al-9Nb alloy decrease due to hydrogen addition at a temperature of 1150 °C and different strain rates. Also the deformation activation energy of the unhydrogenated and hydrogenated alloys was evaluated. According this result, it was established that the main softening mechanism is dynamic recrystallization. Moreover authors show microstructure evolution of the unhydrogenated alloys after high temperature deformation. Despite of interesting results, the presented investigation not relevant to the subject of the journal. In the presented manuscript the Ti-45Al-9Nb alloy was investigated as volume material not in the form of coating. Moreover the methods for analyzing alloys under deformation require using volume materials. Based on this, I recommend sending the manuscript to another journal (Metals, Materials, Journal of Alloys and Compounds etc.).

However, before sending the manuscript should be significantly revised.

Comments to the authors:

(Introduction) Why did you hydrogenate the samples to 0.8 at.%? The reason of choosing hydrogen concentration (0.8 at.%) should be added in the manuscript. What is the solubility limit of hydrogen in the Ti-45Al-9Nb alloy alloy? (Experiment) How was all samples hydrogenated to 0.8 at.%? Is there a variation in concentration in the samples? How did you control the hydrogen concentration in the samples? The answers to these questions should be added in the manuscript. (Results and discussion) The α2, γ and B2 phases are not indicated in the test of the manuscript. (Results and discussion) XRD data is not quantitative analysis. The phase content and lattice parameters from XRD should be calculated. (Figure 4) How did you determine the phase content from EBSD analysis? Is it really possible to detect a phase which content is 0.01%? I am sure that is a mistake. What is the accuracy of the method? These results should be analyzed again. (High-temperature flow behavior of hydrogenated alloy) Figure 5a seems very strange. According to the image, distribution of hydrogen in the hydrogenated samples is nonequilibrium, because the samples destroy in different places (not only working area). It can be happens due to hydrogen accumulation in areas with high strain concentration as a result of wrong sample preparation. Moreover, it is impossible to understand how many samples test for each strain rate. Please pay attention on this result. (Manuscript) The majority results and figures were presented without errors. For example authors show: 76.20%, 23.79% and 0.01% from EBSD analysis; 26.05% (0.0004 s-1), 23.49% (0.001 s-1) and 14.23% (0.0025 s-1) for elongation reduction; 556.94954 KJ/mol for deformation activation energy. The accuracy of the methods does not allow obtaining such values. (Manuscript) The English grammar needs some revision.

Reviewer 2 Report

This paper focuses on the effect of hydrogen on mechanical properties and microstructure evolution of a Ti-45Al-9Nb alloy under uniaxial tension at high temperature. This paper contains some interesting results, but a series of mandatory corrections are suggested in the following:

1) The authors often use the unsuitable English for the technical paper. For example, “So these problems…” (Page 1, Line 38), “But the density of…” (Page 2, Line 48), and “…about 50 s” (Page 3, Line 115). The authors should correct “So”, “But”, and “about” to “However”, “Thus”, and “approximately”, respectively.

2) The authors should correct “1.5418 ng” to “1.5418 Å” (Page 3, Line 121).

3) The authors should correct “.015010” to “3.15010” (Page 10, Line 315).

4) The authors should correct “Figurea 8” to “Figure 8” (Page 10, Line 326).

5) The authors should correct “hydrid” to “hydride” (Page 11, Line 361 and 363).

6) Effect of hydrogen on mechanical properties and microstructure evolution of TiAl alloy under “compressive stress” have been reported (e.g., Refs. 14 and 15). The authors insist that the novelty of the present study is to investigate the effect of hydrogen on mechanical properties and microstructure evolution of TiAl alloy under “tensile stress”. However, there is no discussion about the comparison of compressive and tensile stress in this paper, and the comparison should be discussed in section 3.

7) For the microstructures shown in Fig. 2, the authors describe the common points between (a) and (b), but is there no different points? In addition, the authors should show lower magnification images to describe the common and/or different points between (a) and (b) at macro-scale because the different points at micro-scale are pointed out from the microstructures shown in Fig. 4. For instance, the authors describe that the average size of γ/α2-lamellar colonies is approximately 800 μm in both alloys. However, it seems that the size of γ/α2-lamellar colonies in hydrogenated alloy is smaller than that in unhydrogenated alloy.

8) In the sections of 3.2.1 and 3.2.3, there seems to be little evidence that the dynamic recrystallization is the main reason of decreasing stress with increasing strain. The authors should the evidence by citing previous literatures and/or discussing the relationship between the microstructures shown in Fig. 8 and the stress-strain curves shown in Fig. 5.

9) The authors insist that the reason for the deterioration of high-temperature tensile properties of the hydrogenated alloy was the presence of hydride. However, there is no evidence. Is it difficult to observe the formed hydride by SEM? Moreover, does hydride dissolve at the high-temperature?

10) Figure 9 shows the cracking modes of the hydrogenated alloy deformed at only 1150 °C/0.001 s-1. However, the effects of strain rate and temperature on the cracking modes are not discussed, and the authors should describe it.

Round 2

Reviewer 1 Report

Dear Authors,

The majority of indicated remarks were corrected. Despite of this, the following points should be correct/explane:

I can not understand what is the problem to calculate content of phases if you have diffraction pattern? The authors said that hydrogen increasing the content of α2 phase and B2 phase based on XRD data. To prove this statement, a quantitative XRD analysis should be done. It is impossible to detect the phase which content is 0.01% by EBSD analysis. The content of phases in Ref. [R7, 8] was significantly higher that your case. These results must be analyzed again. Authors neglect the rule of rounding in the text of manuscript. All results containing errors should be correct according this rule.

Reviewer 2 Report

This paper contains some important findings and should be accepted if you would correct the following points.

1) In Figure 6c, the authors should correct "Elonging" to "Elongation".

2) In Figure 6d, the authors should correct "peak stress" to "elongation".

3) The authors should correct “.015010” to “3.15010” (Page 10, Line 329). 

4) The authors described that they also carried out high-temperature compression tests on the same materials. However, effect of hydrogen on mechanical properties and microstructure evolution of other TiAl alloys under compressive stress have been already reported by several authors. At least, the authors should discuss the relationship between the results obtained in the present and previous studies.

Round 3

Reviewer 1 Report

The remarks were corrected. In this form, the manuscript can be accepted for publication.